# Reconstructing Net Primary Productivity in Northern Greater Khingan Range Using Tree Rings

**DOI:** 10.3390/plants14172768

**Published:** 2025-09-04

**Authors:** Yuhang Yang, Yongchun Hua, Qiuliang Zhang, Fei Wang

**Affiliations:** College of Forestry, Inner Mongolia Agricultural University, Hohhot 010000, China; yangyuhang@emails.imau.edu.cn (Y.Y.);

**Keywords:** dendrochronological reconstruction, Greater Khingan Range, drought, climate-growth relationship, tree growth

## Abstract

As critically important global carbon sinks, the net primary productivity (NPP) of boreal forests is crucial for understanding the terrestrial carbon cycle. However, a lack of long-term, high-resolution data has hindered progress in this field. In this study, we used a standardized tree ring chronology of *Larix gmelinii* to identify the dominant factors driving NPP changes in the Northern Greater Khingan Range, applying both Pearson correlation coefficients and SHAP importance values. We then integrated XGBoost and Extreme Random Forest (ERF) models to reconstruct interannual forest NPP across the region from 1968 to 2020. Our results reveal a significant correlation between NPP and tree radial growth, with both processes dominated by growing season drought. The combination of machine learning and tree ring methods proved to be a reliable approach, with the XGBoost model achieving higher reconstruction accuracy than the ERF model. The reconstructed NPP series showed strong regional correlation with MODIS NPP products (r > 0.6) and revealed interdecadal cycles of 10, 28, and 49 years, as well as shorter periodicities of 2–8 and 15–18 years. This study establishes a novel framework for high-resolution NPP reconstruction and clarifies the response mechanisms of the boreal forest carbon cycle to climate change.

## 1. Introduction

Forest ecosystems exhibit the highest global biological productivity and biomass accumulation rates, playing a pivotal role in maintaining regional ecological environments and the global carbon balance [1]. As dynamic carbon pools, they hold irreplaceable carbon and ecological value. Net primary productivity (NPP), a core driver of forest carbon storage, reflects plants’ photosynthetic carbon sequestration capacity while acting as a key element that drives carbon cycles, determines carbon sink scales, and regulates ecological processes [2]. Global warming exerts significant influence on NPP [3,4], and the current lack of long-term, high-precision regional NPP data severely limits the understanding of boreal coniferous forest carbon cycle processes and carbon stock assessments, hindering progress in related research and achievement of “dual-carbon” goals.

Tree rings are direct, continuous indicators of tree radial growth, which inherently document long-term carbon accumulation in forest ecosystems’ aboveground biomass [5,6,7]. These natural “recorders” bridge short-term ecological observations and long-term environmental changes, serving as widely used proxies in climate and environmental research to support studies of past climate variability, ecosystem responses to environmental stress, and historical carbon cycle dynamics [8,9,10]. In dendroclimatology and dendroecology, the existing research has extensively applied multiple tree ring parameters, including ring width indices, wood density, and stable isotopes (δ^13^C, δ^18^O), to analyze climate change patterns and associated ecological responses across diverse spatiotemporal scales [11,12,13,14,15,16]. Tree ring chronologies, for example, have been used to reconstruct historical drought events and climate conditions. These reconstructions extend climate records beyond the instrumental observation period; provide reliable empirical evidence for assessing drought frequency [17], intensity, and trends under climate change; and support regional water resource management and ecological risk assessment [18,19]. Prior studies confirm tree ring chronologies effectively capture interannual and decadal climate variability, reflect seasonal drought impacts on tree growth, establish tree rings’ utility for high-resolution climate reconstruction, and highlight their potential to predict forest ecosystems responses to future climate scenarios [20,21]. Beyond drought reconstruction, researchers have focused on quantifying the sensitivity of tree ring chronologies to key climate drivers, using long-term ring width chronologies, for instance, to evaluate tree growth–climate relationships across scales [22,23]. This work lays a foundation for tree ring use in climate reconstruction while confirming tree rings’ reliability as high-resolution climate proxies and the spatial representativeness of tree ring-based climate indicators [24,25].

Notably, applying tree ring data to NPP reconstruction remains limited. This gap hinders our understanding of long-term forest carbon sink dynamics. Traditional tree ring-based NPP reconstruction methods typically rely on simple linear relationships between ring widths and aboveground biomass increments. While some studies have integrated ring widths with allometric growth models to reconstruct NPP spatiotemporal patterns, these approaches still suffer from low accuracy and weak spatial correlation [26,27]. Moreover, most existing studies focus on large regional scales, creating a scarcity of high-precision, long-term NPP reconstruction research targeting small, ecologically unique regions. This gap limits insights into fine-scale carbon cycle dynamics and their responses to local environmental changes.

The Northern Greater Khingan Range, a vital Northern Hemisphere carbon sink and biodiversity reserve, is highly sensitive to climate warming. Its forest carbon stocks are shaped by interspecific differences in carbon sequestration efficiency [28,29,30]. *Larix gmelinii*, the region’s dominant species, is a key component of this carbon sink. However, its ecosystem functions and carbon sequestration capacity face threats from global warming. Growth differences across altitudinal gradients further modulate regional carbon sink function, making the integration of ring widths with regional NPP dynamics critical to revealing the mechanisms underlying forest growth and carbon cycles [31,32].

To address these research gaps, this study utilizes an autoregressive tree ring chronology of *Larix gmelinii* and integrates tree ring width with regional NPP data, climatic factors, and machine learning approaches—namely XGBoost, ERF, and SHAP analyses. The study aims to (1) identify the climatic factors driving radial growth and NPP variability in the Northern Greater Khingan Range, (2) develop a high-accuracy model to reconstruct regional NPP from 1968 to 2020 and validate it against remote sensing products, and (3) investigate multi-scale correlations and periodic patterns between NPP and tree radial growth.

This work provides a novel methodology for long-term, high-resolution NPP reconstruction in boreal forests, supports accurate assessments of carbon sinks under climate change, and offers scientific guidance for forest management and “dual-carbon” policy formulation.

## 2. Materials and Methods

### 2.1. Study Area Overview

The study area is located in the Chaocha Primeval Forest Region of the Northern Greater Khingan Range in Inner Mongolia (Figure 1), which represents the core area of the northern section of the range, with an elevation of 848 m and geographical coordinates of 50°54′21″ N, 121°30′34″ E. As a critical extension of the Eurasian taiga belt, the Greater Khingan Range constitutes a key component of the Eurasian boreal coniferous forest ecosystems and is recognized as a global hotspot for carbon cycle research [33]. Influenced by both continental monsoon climate and high-latitude cold–humid conditions, the Chaocha Primeval Forest Region has a mean annual temperature of approximately −3 °C and annual precipitation ranging from 450 to 550 mm, with about 70% of precipitation occurring during the growing season (June–August). The seasonal dynamics of temperature and precipitation are illustrated in Figure 2 (Walter–Lieth climate diagram), where the blue curve represents the monthly mean temperature (°C) and the green bars indicate monthly precipitation (mm). The diagram clearly shows cold–dry winters (December–February) and warm–wet growing seasons (June–August).

Forest communities in the region exhibit distinct vertical and horizontal distribution patterns. Larix gmelinii, a cold- and drought-tolerant pioneer species, dominates in both pure and mixed stands, forming typical forest types including Ledum palustre–Larix gmelinii, herb–Larix gmelinii, and riparian–Larix gmelinii forests. Ledum palustre–Larix gmelinii forests are commonly found in poorly drained low-lying wetlands, where the acid tolerance and water-retention capacity of the understory Ledum palustre help maintain the stability of the soil carbon pool [34]. Herb–Larix gmelinii forests occur on relatively gentle slopes, where seasonal herb growth significantly influences understory carbon cycling rates. Riparian–Larix gmelinii forests, supported by abundant water availability in riparian zones, exhibit greater potential for biomass accumulation. An overview of the study area, including sampling points, weather stations, and topographic features, is provided in Figure 1.

### 2.2. Tree Ring Sample Collection and Chronology Construction

To obtain high-resolution records of carbon accumulation, we established four sampling sites in the natural forests of the Chaocha Primeval Forest Region in 2021, following the standardized protocols of the International Tree Ring Data Bank (ITRDB). We maintained a minimum distance of 500 m between adjacent sites to avoid spatial autocorrelation, and *Larix gmelinii* was selected as the target species for sampling. At each site, we extracted 50 cores from 25 randomly selected trees, yielding a total of 200 cores from 100 trees. The sampled trees met the following criteria: (1) no visible signs of pest infestation, disease, or mechanical damage; (2) vigorous growth; (3) no evidence of human disturbance in the immediate vicinity. For each tree, two cores were extracted at breast height (1.3 m) along the east–west and north–south axes to mitigate the influence of asymmetric light and water availability on radial growth. Drill holes were subsequently sealed with medical-grade petrolatum to prevent infection and water loss. As this study focused on tree ring width—a robust annual growth proxy—we did not analyze the earlywood/latewood ratio. We employed skeleton plotting solely to assist in ring width measurements. Measurements were performed using a LINTAB 6.0 tree ring width analyzer with a precision rate of 0.001 mm. The COFECHA program was used to identify and correct measurement errors and to screen for cores with well-preserved climatic signals and long chronologies. We applied a negative exponential detrending function using the ARSTAN program to generate an autoregressive tree ring width index chronology. Following this, we performed denoising to exclude non-climatic trends. Parameters were tuned and optimized based on the statistical characteristics of the output chronology to select the optimal chronology representing the regional growth patterns of most trees. The final autoregressive chronology was built from 72 cores from 36 trees (selected from the initial 100 trees). All 72 cores exhibited high data integrity and strong chronology statistics. This chronology was chosen as it effectively removes tree-specific growth trends while preserving interannual variability driven by the climate, thereby providing a reliable proxy for carbon accumulation to support net primary production (NPP) reconstruction.

### 2.3. Acquisition of Modeling Data

#### 2.3.1. MODIS NPP Data

For long-term, high-precision NPP reconstruction, the Net Primary Productivity (NPP) data used in this study were obtained from the MOD17A3HGF V6 product, a joint release by the National Aeronautics and Space Administration (NASA) and the United States Geological Survey (USGS) (https://lpdaac.usgs.gov/products/mod17a3hgfv006/ (accessed on 26 August 2025)). Derived from the inversion of Moderate Resolution Imaging Spectroradiometer (MODIS) remote sensing data, this product provides annual cumulative NPP values with a spatial resolution of 500 meters and a Sinusoidal projection. For this research, we used 500 m spatial resolution Net Primary Productivity (NPP) data of the Northern Greater Khingan Range. selected the corresponding NPP image data and extracted the 2000–2020 dataset to serve as baseline values for model validation.

#### 2.3.2. Meteorological Data

To clarify how climatic factors drive forest carbon fixation, we obtained basic climate data (mean temperature and total precipitation) from 1968 to 2020 from the Genhe Meteorological Station, the closest station to our sampling sites. We also incorporated two key hydrometeorological metrics: the Palmer Drought Severity Index (PDSI), which effectively reflects growing season water stress on tree carbon assimilation; and the vapor pressure deficit (VPD), which quantifies atmospheric aridity and reveals the coupling between stomatal conductance and photosynthetic carbon fixation. PDSI data were sourced from the 0.5° × 0.5° CRU dataset provided by the Royal Netherlands Meteorological Institute (http://climexp.knmi.nl). VPD data were obtained from the TerraClimat dataset (http://www.example.com).

### 2.4. Statistical Methods

To address the limitations of traditional NPP reconstruction methods, we employed a multi-faceted analytical approach. First, we used Pearson correlation analysis (*p* < 0.05) to identify climate factors influencing NPP. We then integrated SHAP (SHapley Additive exPlanations) value analysis to quantify the contribution of each factor, thereby overcoming the inability of correlation analysis to capture non-linear relationships. Furthermore, we introduced the SHAP interpretable machine learning framework to elucidate the mechanisms by which these driving factors change over time.

For the reconstruction process, we utilized both the Extreme Random Trees (ERF) and eXtreme Gradient Boosting (XGBoost) models. The tree ring width index (RWI) and selected climate factors served as input variables to reconstruct the NPP series from 1968 to 2020. The XGBoost model, which is optimized through gradient boosting, offers superior performance in processing high-dimensional, non-linear data.

To validate the models and analyze dynamics, we evaluated the model reliability using the coefficient of determination (R^2^) and the root mean square error (RMSE). The model outputs were combined with SHAP results to clarify the influence of each factor on NPP changes. A wavelet analysis was applied to identify multi-timescale cycles in the NPP series. Finally, we used the Mann–Kendall mutation test and spatial autocorrelation analysis to identify historical evolution patterns of carbon sink capacity and their spatial consistency with MODIS NPP data.

## 3. Results and Analysis

### 3.1. Statistics of Chronology Characteristic Values

The quality of the tree ring chronology is fundamental to reliable NPP reconstruction. A statistical analysis of the *Larix gmelinii* autoregressive chronology (Table 1, Figure 3) yielded two key findings: the interseries correlation coefficient of the standard chronology was 0.43; the first principal component explained 43.03% of the variance. These results indicate uniform growth trends among the samples, confirming that the chronology effectively reflects average regional tree growth patterns [35]. The chronology also exhibited strong performance across key quality metrics, including an overall expressed population signal (EPS) of 0.973, a mean sensitivity of 0.182, and a signal-to-noise ratio (SNR) of 32.971. These indicators confirm the chronology’s high capacity to capture climatic signals, thereby providing a robust data foundation for subsequent NPP reconstruction [36].

### 3.2. Statistics of NPP Changes in the Study Area

Analyzing existing NPP records is critical for guiding regional reconstruction efforts. In the study area, NPP showed a significant increasing trend (linear regression slope = 6.004, *p* < 0.05), although a notable decline occurred between 2006 and 2009. Over the past two decades, regional ecosystem productivity has increased at an average annual rate of approximately 6 units. This trend is consistent with the enhanced carbon sink capacity observed in other high-latitude regions under global climate change, thereby supporting efforts to achieve “dual-carbon” goals. The interannual variations in observed NPP (2000–2020) and the linear regression fit are presented in Figure 4.

### 3.3. Effects of Climate Factors on Larix gmelinii Radial Growth and Regional NPP Changes

A correlation analysis was performed to identify key predictor variables for modeling. Regional NPP changes were strongly correlated with *Larix gmelinii* radial growth (r = 0.57, *p* < 0.05; Figure 5a). The NPP series was also strongly correlated with annual precipitation and the Palmer Drought Severity Index (PDSI), although no significant correlation was observed between the tree ring width index (RWI) and these two factors.

In contrast, the RWI series showed significant correlations with the PDSI (June of the current year), mean temperature (May and December), mean maximum temperature (December), mean minimum temperature (May, August, December), and relative humidity (November) (Figure 5b,d).

The NPP series exhibited significant correlations with the PDSI (June–December of the current year), mean maximum temperature (August), and relative humidity (January, February, June, August, September–December).

In the study area, *Larix gmelinii* radial growth was primarily limited by mid-growing-season drought and mean temperature. In contrast, NPP changes were strongly governed by precipitation, growing-season drought, and tree ring width, identifying drought and precipitation as the key constraints on NPP. Based on these correlation patterns, RWI, PDSI, and precipitation (P) were selected as input variables for the model.

### 3.4. NPP Reconstruction and Accuracy Validation

Building on these correlation patterns, we employed a SHAP value analysis to further refine the selection of climate factors, ultimately confirming the tree ring width index (RWI), Palmer Drought Severity Index (PDSI), and precipitation (P) as the final predictor variables for modeling. The contribution of each factor to the NPP model, quantified by mean SHAP importance values, and the directional impact of factor changes on the model output, represented by individual SHAP values, are visualized in Figure 6.

We then applied both the Extreme Random Forest (ERF) and eXtreme Gradient Boosting (XGBoost) models to reconstruct the NPP series from 1968 to 2020. The XGBoost model, which is optimized through gradient boosting, offers superior performance for processing high-dimensional, non-linear data (Figure 7).

Model reliability was evaluated using the coefficient of determination (R^2^) and the root mean square error (RMSE) (Table 2). A direct accuracy comparison between the NPP series reconstructed by the XGBoost and ERF models is presented in Figure 6. The XGBoost model achieved a higher variance explanation rate (78%) than the ERF model (62%). The superior performance of the XGBoost model was further confirmed by its lower mean squared error (MSE = 472.49), RMSE (21.74), and mean absolute error (MAE = 10.13). In contrast, the ERF model exhibited a more favorable Durbin–Watson statistic (1.25), indicating better handling of autocorrelation in the residuals.

### 3.5. Historical NPP Variation Trends in the Study Area

Historical NPP changes showed a significant increasing trend (Figure 8). Both model-reconstructed NPP series were consistent with the measured NPP series, with a notable decline during 2006–2009. The ERF-reconstructed NPP series showed infrequent fluctuations, with obvious declines in 1971, 1986, 1990, 1998, 2002, and 2007; a growth period during 1974–1984; and a decline period during 1988–2002. The XGBoost-reconstructed series showed frequent fluctuations; an increasing trend with growth–decline cycles; and obvious declines in 1971, 1977, 1985, 2000, and 2005.

### 3.6. Spatiotemporal Characteristics of Reconstructed NPP Series

#### 3.6.1. Mutation Characteristics of Reconstructed NPP Series

To deepen our understanding of regional carbon sink dynamics, we applied sliding *t*-tests and Mann–Kendall (MK) mutation tests to analyze the mutation characteristics of Net Primary Productivity (NPP) and ring width index (RWI) series, with results presented in Figure 9. Figure 9a shows sliding *t*-test and MK mutation analysis results for the observed NPP series (2000–2020), Figure 9b shows the same for the RWI series (1968–2020), and Figure 9c for the XGBoost-reconstructed NPP series (1968–2020); this series exhibited significant mean mutations (*p* < 0.05) in 2001–2006, 2011, and 2013–2014, consistent with sliding *t*-test results that identified significant mean mutations in 2001, 2002, 2003, 2004, 2005, 2006, 2011, 2013, and 2014 (*p* < 0.05). Figure 9d shows sliding *t*-test and MK mutation analysis results for the ERF-reconstructed NPP series (1968–2020); The sliding t-test revealed mutations in the reconstructed sequence in 1983 and 2002 (*p* < 0.05). For the observed NPP series, sliding *t*-tests also revealed significant mean mutations in 2002, 2003, and 2011 (*p* < 0.05). Across the entire study period, MK tests indicated a significant upward trend (*p* < 0.05); the forward test statistic (UF) remained consistently above the 0.05 significance threshold, with no intersection between UF and the backward test statistic (UB), confirming the absence of mutation points.

Abrupt NPP changes in the Northern Greater Khingan Range are shaped by the interactive effects of climate change and local human disturbances, with regional NPP dynamic variation characteristics highly consistent with regional tree radial growth patterns (Figure 9). Climate change acts as the primary driver of abrupt NPP changes by regulating hydrothermal conditions, with the Palmer Drought Severity Index (PDSI) and RWI emerging as key factors influencing regional NPP variations. Specifically, the PDSI modulates the photosynthetic efficiency and radial growth of *Larix gmelinii*-dominated coniferous forests, in turn inducing abrupt decreases in regional NPP.

Human activities have emerged as a significant disturbance factor that disrupts the climate-driven unidirectional trend of NPP, leading to abrupt local NPP changes [37]. For instance, in the Northern Greater Khingan Range, overgrazing (due to rising livestock numbers) and vegetation destruction from agricultural expansion have not only caused an overall decline in NPP but also triggered abrupt NPP changes in the affected areas [25,38,39].

#### 3.6.2. Periodic Variations of Reconstructed NPP Series

A wavelet analysis was applied to verify the reliability of reconstructed NPP series, clarify historical NPP changes in the Northern Greater Khingan Range, and reveal long-term climate oscillation impacts on regional carbon cycles. The wavelet real part analysis showed that the ERF-reconstructed NPP, XGBoost-reconstructed NPP, and RWI shared identical main cycles and similar variation patterns (Figure 10)—exhibiting respective main cycles of 10 a, 29 a, and 50 a during 1980–2020, alongside significant cycles of 2–8 a, 15–18 a, and 20–35 a. The wavelet coherence analysis identified significant positive coherence between ERF NPP, XGBoost NPP, and RWI series at the 4–8 a scale during 2000–2020 (Figure 11), confirming synchronous positive changes across these datasets. ERF NPP also displayed an additional 0–4 a coherence period during 1985–2008, with ERF NPP leading RWI changes; wavelet period comparisons further revealed two common 0–4 a cycles across all series during 1996–2020.

#### 3.6.3. Spatial Correlation of Reconstructed NPP Series

To assess the regional representativeness of the reconstructed NPP series, we conducted a spatial correlation analysis between the 1968–2020 reconstructed NPP data and MODIS NPP data (Figure 12). The study area characterized as a typical forest type with climate driving mechanisms consistent with surrounding regions exhibited strong spatial correlation between the reconstructed NPP series and MODIS NPP data across the Northern Greater Khingan Range, as well as adjacent areas of Mongolia and Russia, with correlation strengths increasing near the study area itself. This pattern confirms the reconstructed NPP series is suitable for both small-scale local analyses and large-scale regional assessments.

## 4. Discussion

Drawing on the characteristics of NPP reconstruction models and spatiotemporal patterns of NPP (derived from mutation tests, wavelet analyses, and spatial correlation analyses), this study discusses methodological limitations, clarifies the link between *Larix gmelinii* radial growth and regional carbon sinks, and offers support for local “dual-carbon” initiatives. It further proposes methods and a model framework for dynamic monitoring of forest ecosystem NPP; analyzes regional carbon sink response mechanisms; and provides theoretical and technical support for scientific forest management, accurate carbon sink assessment, and emission reduction practices under “dual-carbon” goals.

### 4.1. Mutation Characteristics and Limitations of Reconstructed NPP

Sliding window *t*-tests and Mann–Kendall (MK) trend tests were applied to the NPP RWI XGBoost-reconstructed NPP and ERF-reconstructed NPP series. For the NPP and RWI series, sliding window *t*-tests identified mutation time points where test statistics exceeded the 95% confidence threshold, indicating potential abrupt changes in the datasets. MK trend tests showed NPP forward test statistics (UF blue line) remained consistently above the critical value, confirming a long-term significant upward trend. However, this strong trend may mask short-term cycles when paired with periodic analyses (e.g., fluctuations revealed by the ERF model), a pattern consistent with the known limitation that MK tests are insensitive to periodic changes and short-term climate fluctuations under strong directional trends, highlighting the challenge of using single methods to characterize complex ecosystem dynamics.

As a direct indicator of tree radial growth, the RWI exhibited an upward trend in MK tests but with more pronounced fluctuations differing from NPP trends and reflecting distinct scale-specific characteristics in how individual tree growth and regional productivity respond to climate and environmental changes.

For the model-reconstructed NPP series, the XGBoost model sensitive to short-term fluctuations frequently crossed the critical value in sliding window *t*-tests, demonstrating strong ability to capture NPP mutations driven by extreme climate events. Its MK trend test showed large fluctuations in UF statistics (Figure 9c), indicating poor stability in simulating long-term trends and highlighting XGBoost’s limitation in quantifying climate factor contributions and analyzing the intrinsic mechanisms of ecosystem responses to climate change; under “dual-carbon” goals, this hinders accurate assessments of carbon sink responses to climate factors and their associated feedbacks.

The ERF model by contrast maintained stability in quantifying factor importance and simulating long-term trends but lacked capacity to fit nonlinear climate responses. Under climate warming, it failed to accurately simulate NPP changes driven by complex nonlinear interactions between precipitation temperature and other factors (e.g., synergistic effects of growing season drought and warming), compromising the accuracy of carbon sink estimations [38,39].

Future model optimization efforts should center on multi-method integration leveraging XGBoost’s strength in capturing short-term dynamics, combining ERF’s stability in simulating long-term trends and integrating interpretable machine learning tools such as SHAP to quantify climate factor contributions and analyze nonlinear response mechanisms. Incorporating additional ecological process parameters such as vegetation phenology and soil moisture dynamics will lead to further enhanced model performance in simulating complex ecosystems, ultimately providing technical support for accurate assessment and prediction of forest carbon sinks via NPP dynamics under “dual-carbon” goals.

### 4.2. Dominant Factors for Larix gmelinii Radial Growth and Regional NPP Changes in the Study Area

This study identified the June PDSI as a significant driver of *Larix gmelinii* radial growth (*p* < 0.01), suggesting that under climate warming, growing season drought may increasingly dominate radial growth in the study area. This conclusion is supported by the correlation analysis (Figure 5b), which shows that the tree ring width index (RWI)—a proxy for tree radial growth—exhibits a significant positive correlation with the June PDSI, confirming that wetter growing seasons promote radial growth. This relationship occurs because water availability during the mid-growing season (June–July) is critical for tree growth and directly regulates radial growth processes [40,41,42,43]. Concurrent correlation analyses further revealed a strengthened correlation between precipitation and tree growth, providing additional evidence that water supply is the primary limiting factor for *Larix gmelinii* growth in this area [44].

With climate warming, the limiting factors for *Larix gmelinii* growth have shifted. The mean temperature during the mid-growing season showed a significant positive correlation with radial growth, indicating that moderate warming may promote growth. However, the enhanced correlation between precipitation and growth reflects the rising importance of water supply, which is consistent with previous studies [45,46,47,48]. As the peak growing month, June experiences high temperatures that intensify transpiration; when insufficient precipitation occurs, this leads to severe drought stress, which reduces radial growth (resulting in narrow rings) and weakens the carbon sink capacity of *Larix gmelinii*. This aligns with findings that tree rings effectively capture the impact of climate extremes (e.g., droughts) on radial growth, even in young tropical trees, confirming the universality of tree ring responses to climate extremes across biomes [49,50].

Climate factors influenced *Larix gmelinii* radial growth and regional net primary productivity (NPP) in distinct ways (Figure 5a). The tree ring width index (RWI), a proxy for radial growth, was primarily correlated with the June PDSI and specific monthly temperatures (e.g., May mean temperature). In contrast, regional NPP correlated with a broader PDSI window (June–December) and humidity (e.g., August humidity), highlighting the differing response scales of individual tree growth versus regional ecosystem productivity. Specifically, *Larix gmelinii* radial growth was strongly affected by the June PDSI and current-year temperatures—including mean temperatures in May and December; the December mean maximum temperature; and mean minimum temperatures in May, August, and December (Figure 5c). Regional NPP, by contrast, exhibited greater sensitivity to air humidity and longer-term drought conditions. This discrepancy likely arises because regional NPP is heavily regulated by the total biomass of the arboreal layer [51]. As *Larix gmelinii* dominates the aboveground biomass in the arbor-dominated forests of the Northern Greater Khingan Range, its radial growth strongly influences this pool, but regional NPP integrates more components and responds to a wider range of climate variables. Under ongoing global warming, both NPP and *Larix gmelinii* radial growth in the study area are shifting from temperature limitation to water limitation [52]. Consequently, the increased frequency and intensity of heatwaves and droughts are likely to become critical factors shaping future forest growth and regional carbon sequestration.

### 4.3. Analysis of NPP Variation Characteristics in the Study Area

From 1968 to 2020, NPP in the Northern Greater Khingan Range exhibited a significant increasing trend, indicating an enhancement of the regional carbon sink capacity. However, the reconstructed NPP series showed considerable interannual fluctuations, likely driven by divergent changes across sub-regions. For instance, while NPP growth was insignificant in some areas, Genhe saw significant increases, resulting in these frequent fluctuations at the regional scale. Between 2006–2009 and 2000–2018, NPP changes were strongly shaped by regional factors; ecological protection measures (e.g., grazing prohibition) in fragile areas such as the Xin Barag Left Banner curbed desertification and boosted regional NPP [53], whereas human disturbances (e.g., excessive logging, irrational land use) reduced NPP in other areas [54,55].

The SHAP analysis revealed that regional NPP changes were significantly influenced by both climatic factors (primarily the Palmer Drought Severity Index, PDSI, and precipitation) and tree radial growth (as measured by the tree ring width index (RWI); Figure 6a). Both the PDSI and RWI exhibited positive correlations with NPP, collectively driving the significant increase in the regional carbon sink capacity. This relationship exists because the RWI of *Larix gmelinii*—the dominant species—directly reflects aboveground carbon accumulation, which is a major component of regional NPP. These findings confirm that drought (mediated by PDSI) exerts a more pronounced impact on carbon sequestration than other factors, a conclusion aligned with previous research establishing that mountain NPP variations are jointly driven by climate and tree growth [56].

The wavelet analysis further identified multi-timescale cycles (e.g., 10, 28, and 49 year) in the NPP series. This provides strong evidence that the climate plays a dominant role in these dynamics and indicates that the boreal forest carbon sink is regulated by large-scale global climate systems. However, long-term predictions of regional carbon sinks remain uncertain, necessitating additional research to support “dual-carbon” goals. Future work should prioritize the cross-time-scale interactions between climate and biological factors, alongside the long-term impacts of human activities, to improve predictions of boreal forest carbon sink dynamics.

## 5. Conclusions

This study presents a comprehensive analysis of *Larix gmelinii* growth and net primary productivity (NPP) dynamics in the Greater Khingan Range. We developed a standard tree ring chronology, analyzed its correlations with climatic factors and regional NPP, reconstructed historical NPP data with higher accuracy than traditional methods, and performed a wavelet analysis on the tree ring data. The key conclusions are as follows:

(1) Growing season drought is the dominant factor limiting both *Larix gmelinii* radial growth and regional NPP. Regional NPP dynamics are strongly controlled by tree radial growth and drought conditions.

(2) Over the past two decades, NPP in the study area has exhibited a significant upward trend, although historical reconstructions reveal substantial fluctuations. Both machine learning models—Extreme Random Forests (ERF) and XGBoost—achieved robust reconstruction results, with the reconstructed NPP series demonstrating strong spatial representativeness.

(3) The wavelet analysis revealed consistent patterns between the tree ring width and reconstructed NPP series across the 2–8 a and 30–45 a cycles, sharing significant common periodicities and exhibiting synchronous positive changes. Furthermore, the two machine learning models exhibited contrasting strengths and limitations; the XGBoost model delivered higher reconstruction accuracy (determination coefficient R^2^ = 0.78; RMSE = 21.74) but exhibited reduced stability in simulating long-term trends (as evidenced by fluctuations in the UF statistic; Figure 9c). In contrast, the ERF model demonstrated greater long-term trend stability (Durbin–Watson statistic = 1.25) but a weaker capacity to capture nonlinear climate responses, such as drought-warming synergies. Future research should integrate the strengths of both models to enhance overall performance.

## Figures and Tables

**Figure 1 plants-14-02768-f001:**
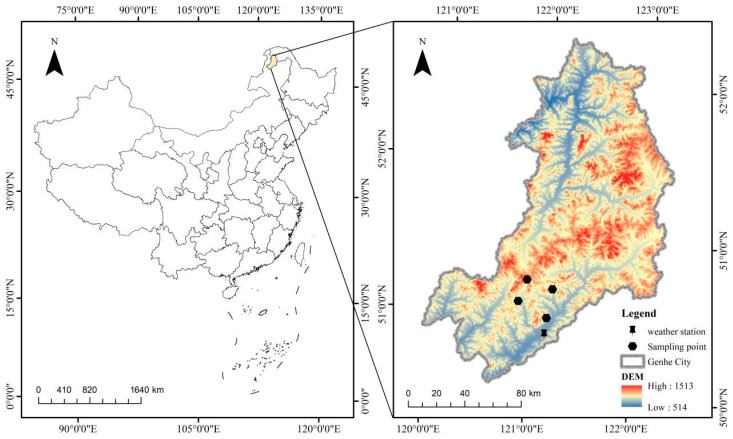
Overview of the study area.

**Figure 2 plants-14-02768-f002:**
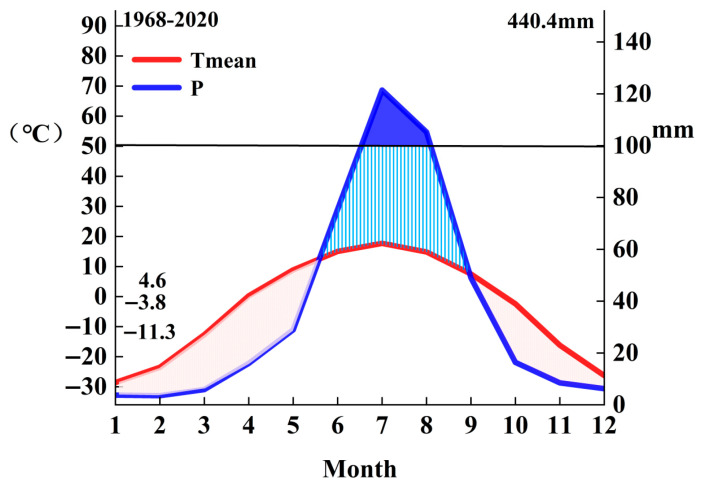
Walter–Lieth diagram of the study area.

**Figure 3 plants-14-02768-f003:**
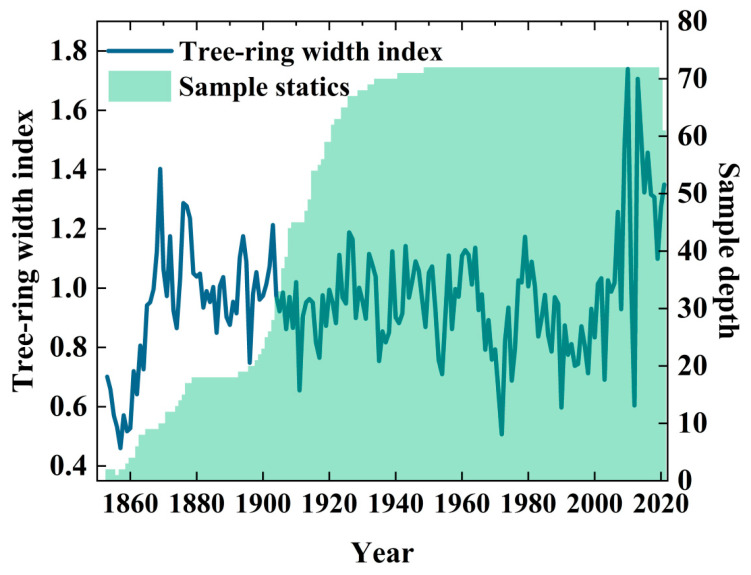
Statistics of *Larix gmelinii* ring widths.

**Figure 4 plants-14-02768-f004:**
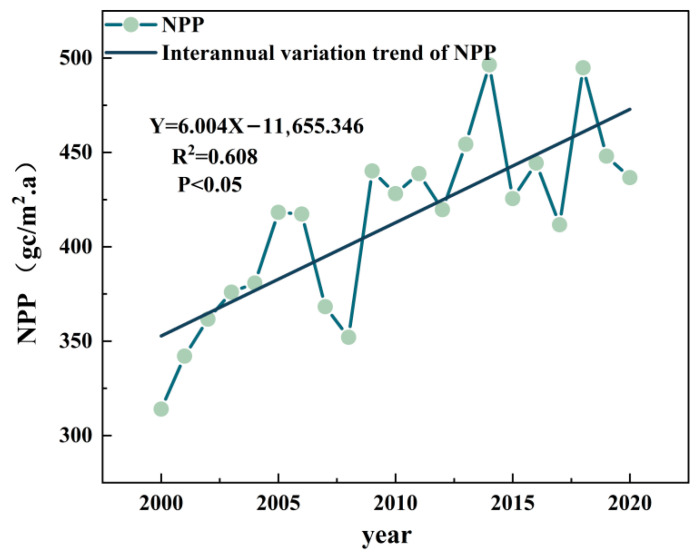
Interannual variations in NPP in Northern Greater Khingan Range.

**Figure 5 plants-14-02768-f005:**
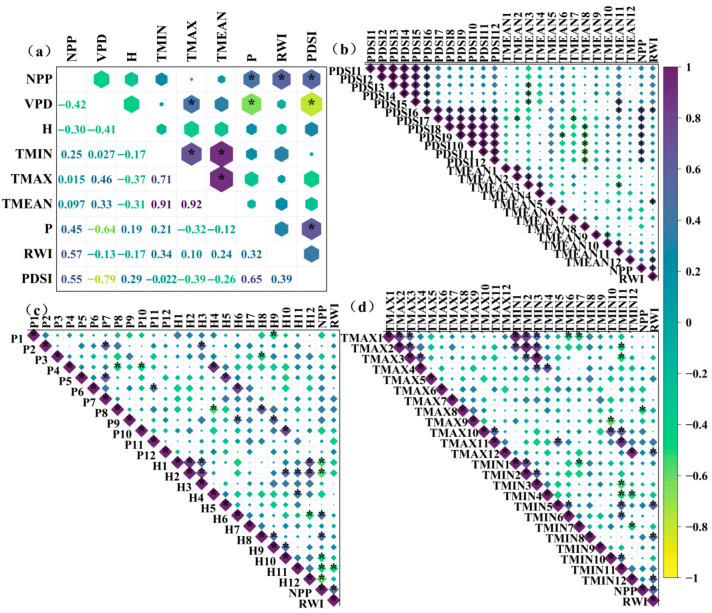
Correlation diagrams of factors and monthly correlations: (**a**) correlations among factors; (**b**) correlations of NPP and RWI with PDSI and monthly mean temperature (TMEAN); (**c**) correlations of NPP and RWI with precipitation (P) and monthly mean humidity (H); (**d**) correlations of NPP and RWI with monthly mean maximum temperature (TMAX) and monthly mean minimum temperature (TMIN). * indicates *p* < 0.05.

**Figure 6 plants-14-02768-f006:**
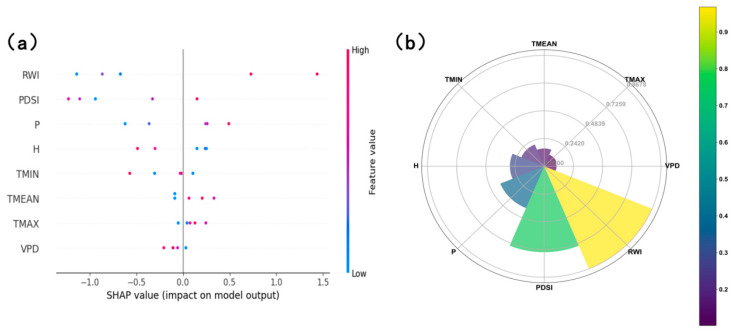
The SHAP importance value analysis of various factors and NPP: (**a**) SHAP importance value forest plot, where bar length represents each factor’s average absolute SHAP value (quantifying its contribution to NPP) and factors are ordered by importance; (**b**) SHAP value fan plot, where a positive SHAP value indicate that a factor’s current value promotes NPP.

**Figure 7 plants-14-02768-f007:**
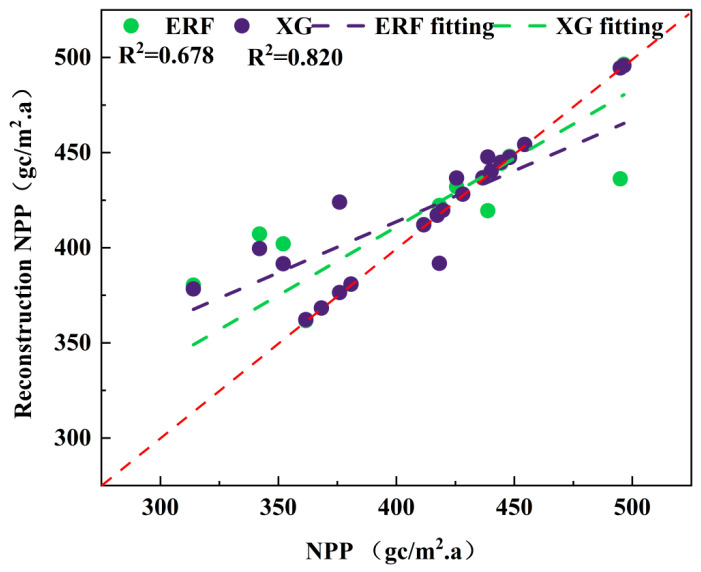
Accuracy comparison of NPP series reconstructed by XGBoost and ERF models. The red dotted line represents the 1:1 line.

**Figure 8 plants-14-02768-f008:**
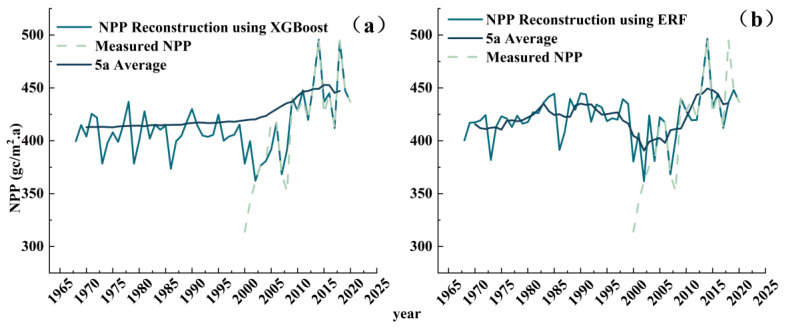
Periodic variations of reconstructed NPP series: (**a**) ERF model-reconstructed NPP (1968–2020) with marked decline years (1971, 1986, 1990, 1998, 2002, 2007); (**b**) XGBoost model-reconstructed NPP (1968–2020) with marked decline years (1971, 1977, 1985, 2000, 2005).

**Figure 9 plants-14-02768-f009:**
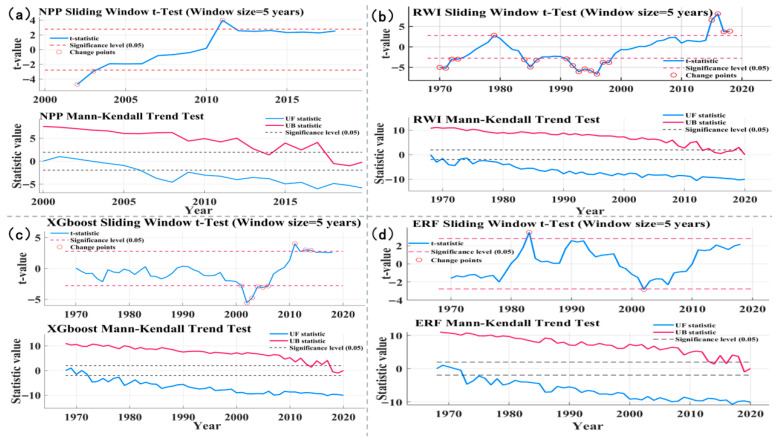
Mutation tests of NPP series and RWI series: (**a**) sliding *t*-test and MK mutation analyses of NPP series; (**b**) sliding *t*-test and MK mutation analyses of RWI series; (**c**) sliding *t*-test and MK mutation analyses of XGBoost NPP series; (**d**) sliding *t*-test and MK mutation analyses of ERF NPP series.

**Figure 10 plants-14-02768-f010:**
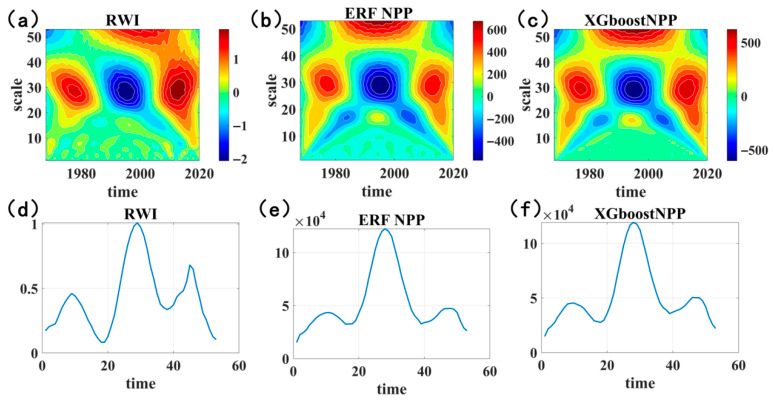
Wavelet real parts and main cycles of reconstructed NPP series and tree ring width series (RWI): (**a**) wavelet real part of RWI; (**b**) wavelet real part of ERF-reconstructed NPP; (**c**) wavelet real part of XGBoost-reconstructed NPP; (**d**) main cycles of RWI wavelet real part; (**e**) main cycles of ERF-reconstructed NPP wavelet real part; (**f**) main cycles of XGBoost-reconstructed NPP wavelet real part.

**Figure 11 plants-14-02768-f011:**
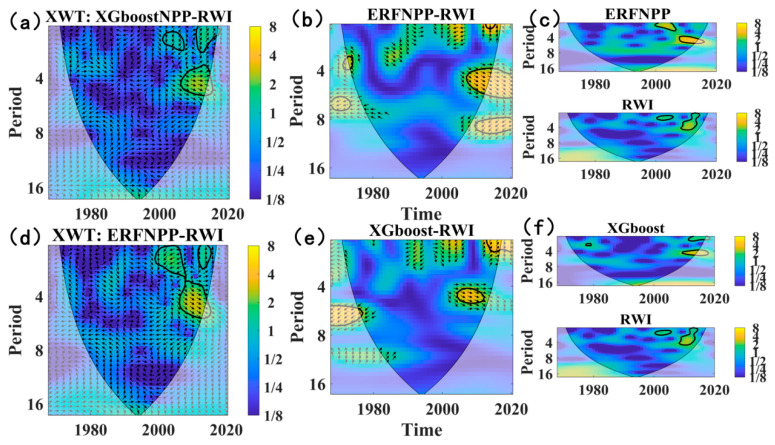
Wavelet coherence diagrams of reconstructed NPP series and tree ring width series (RWI): (**a**) the XWT wavelet analysis of ERF NPP and RWI series; (**b**) the wavelet analysis of ERF NPP and RWI series; (**c**) wavelet coherence cycles of ERF NPP and RWI series; (**d**) the XWT wavelet analysis of XGBoost NPP and RWI series; (**e**) the wavelet analysis of XGBoost NPP and RWI series; (**f**) the wavelet coherence cycles of XGBoost NPP and RWI series. The black area represents the significant coherence interval.

**Figure 12 plants-14-02768-f012:**
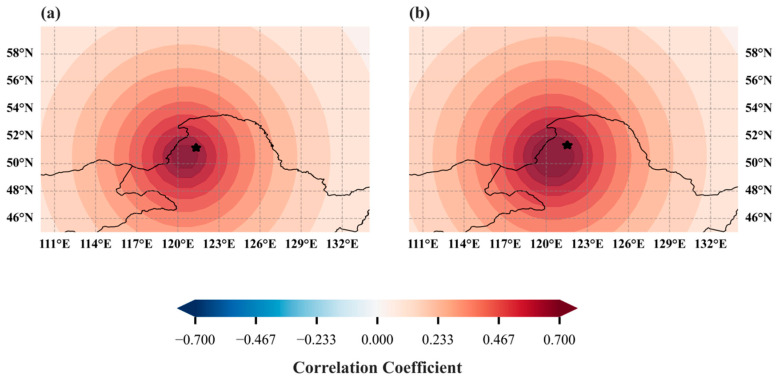
Spatial correlation between model-reconstructed NPP series and MODIS NPP series: (**a**) spatial correlation between ERF-reconstructed NPP and MODIS NPP series; (**b**) spatial correlation between XGBoost-reconstructed NPP and MODIS NPP series. The pentagram indicates the approximate position of the study region.

**Table 1 plants-14-02768-t001:** Statistics of chronology characteristics.

Year	Number of Cores	Average Series Length	Characteristic Values of Standard Chronology			Statistical Indicators of Common Interval				
			MS	SD	AC	SNR	EPS	VF	Overall Sample Representativeness	Inter-Tree Correlation Coefficient
1855–2021	72	166	0.1823	0.1835	0.3460	32.917	0.967	43.03%	0.973	0.43

**Table 2 plants-14-02768-t002:** Parameters of reconstruction models.

	R^2^	MSE	RMSE	MAE	F	DW
ERF	0.62	825.27	28.73	15.17	22.57	1.25
XG	0.78	472.49	21.74	10.13	57.25	0.83

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
