# Peer review of "Reconstructing Net Primary Productivity in Northern Greater Khingan Range Using Tree Rings"

_plants, 2025, doi:10.3390/plants14172768_

Round 1

Reviewer 1 Report

Comments and Suggestions for Authors

Review

of manuscript “Reconstructing Net Primary Productivity in Northern Greater Khingan Range Using Tree Rings” submitted to the Plants

(Manuscript Number “plants-3829173”)

General comments:

The study provides a new and reliable method for assessing the dynamics of carbon sinks in boreal forests by combining traditional tree-ring data with modern machine learning techniques. This is particularly important for understanding the effects of climate change on key species like the Dahurian larch. The results confirm that drought during the growing season is the primary limiting factor for the growth of these trees, a critical finding in the context of global warming. By accurately reconstructing historical NPP, the research allows for a better understanding of the ecosystem's response mechanisms to climate change. Ultimately, the data and models developed provide essential theoretical and technical support for achieving the “dual-carbon” goals, enabling more precise forest resource management and a more accurate prediction of their carbon sequestration capacity. For these reasons, I find the manuscript interesting, but I have some minor recommendations, as listed below:

L 23: It would be better if the keywords do not duplicate the title. This will ensure better "visibility" of the publication during searches.

L 37: Use quotation marks for "dual-carbon," as well as in other parts of the text (e.g., line 160).

L 60: When MODIS is first mentioned or in section 2.3.1, it is appropriate to state the full name: Moderate Resolution Imaging Spectroradiometer.

L 71: Add Figure 1 to the text.

L 68-88: Given that there is a climate station in the region, it would be a good idea to supplement the climate description with Walter-Lieth diagrams.

L 93-95: If I understand correctly, samples were taken from 36 trees from 4 sites, i.e., 9 trees per site. Is this correct? If so, it would be good to explain this.

L 143: Add Figure 2 to the text.

L 153: In Figure 2, the label "Sample statistics" should be replaced with "Sample depth." Both terms – "sample statistics" and "sample depth" – refer to the number of samples but have different meanings and are used for different purposes in dendrochronology. Sample depth refers directly to the number of tree-ring samples used for a given year to build a chronology. Sample statistics is a more general term that describes the set of statistical indicators calculated from the samples, i.e., your table 1.

L 157: Add Figure 3 to the text.

L 163: Larix gmelinii should be in italics, even if the entire style is italics (which may be why it is not).

L 184-188: Add Figures 5 and 6 to the text.

L 193: Add Table 2 to the text.

L 196: The colours in Figure 5 should have clearer boundaries; currently, the colours of the "fitting line" are too similar.

L 198: Supplement the figure description. In the current version, it is not clear what (a) and (b) represent.

L 208: Same comment as for Figure 6.

L 211-218: Add Figure 8 with the corresponding labels for greater clarity.

Between line 253-254: For Figure 11, it would be better to show a larger spatial scope. Currently, it is too small, and to visualize the spatial context, one has to compare it with Figure 1.

L 271: There is no comment on these mutations – what they are caused by what they show, or what they hide. These sudden significant changes in the behaviour of the data series could be due to changes in climate, human activity, or natural disturbances, but there is no comment on this.

L 272: Which figure? It appears that figures are often omitted from the text when they are being commented on.

L 284: "caused by extreme climate events"—is it only due to this? See the comment for line 271.

L 304: Same comment as for line 163.

L 309-311: The text would benefit if this was visualized in some way, i.e., add a figure name.

L 312-314: Same comment.

L 321: This repeats line 305.

L 321: "specific monthly temperatures" – which ones exactly?

L 345-346: Tree-ring width is a result of climate factors, so NPP cannot have been influenced by climate factors and radial growth. Rather, NPP is influenced by climate factors, and this relationship is "visualized" through radial growth.

L 348: "improved climate conditions" – okay, temperatures are increasing, but are the amount of precipitation and atmospheric moisture also increasing? Above, you commented on the limiting effect of moisture, especially during the growing season. Consider whether these are truly "improved climate conditions."

L 366-368: In the main text, you point out the positive and negative aspects of both models; it would be good to mention them here in the conclusion as well.

Others remark:

  • different citation styles are noted (e.g., line 30 vs 37, as well as in other places);
  • two references were not found in the reference list (Meng S W et al., 2017 and Weng Y T et al., 2021);
  • one reference from the reference list was not found in the main text:

Dietrich, V., Skiadaresis, G., Schnabel, F., et al. (2024). Identifying the impact of climate extremes on radial growth in young tropical trees: A comparison of inventory and tree-ring based estimates. Dendrochronologia, 86, 126237.https://doi.org/10.1016/j.dendro.2024.126237

I believe my comments will provide valuable insights to help the authors refine and strengthen the manuscript. I look forward to reading the revised version.

Author Response

Thank you very much for taking the time to review this manuscript.  Your positive evaluation of our study and constructive suggestions have been of great help to us.  We have carefully addressed each of your recommendations, and the corresponding revisions/corrections have been highlighted in the re-submitted files.  Please find the detailed responses below. For specific information on image modification, please refer to the attachment

Comments 1 (Q1, L 23): It would be better if the keywords do not duplicate the title. This will ensure better "visibility" of the publication during searches.

Response 1: Thank you for pointing this out. We agree with this comment. Therefore,we have revised the keywords to avoid duplication with the title. The title of the manuscript is "Reconstructing Net Primary Productivity in Northern Greater Khingan Range Using Tree Rings", and the revised keywords are "Dendrochronological reconstruction;Greater Khingan Range;drought;Climate-growth relationship;tree growth", which do not repeat the content of the title. This revision improves the "visibility" of the publication in search engines. The revised keywords can be found in the "Keywords" section of the revised manuscript.

Comments 2 (Q2, L37): Use quotation marks for "dual-carbon," as well as in other parts of the text (e.g., line 160).

Response 2: Agree.Thank you for pointing this out. We have added quotation marks for all instances of "dual-carbon" in the text. For example, the original content at L37 ("hindering research and achievement of dual-carbon goals") has been revised to "hindering research and achievement of 'dual-carbon' goals", and the content at L160 ("which is significant for achieving dual-carbon goals") has been revised to "which is significant for achieving 'dual-carbon' goals". All revisions of "dual-carbon" with quotation marks are applied throughout the entire revised manuscript.

Comments 3 (Q3, L 60): When MODIS is first mentioned or in section 2.3.1, it is appropriate to state the full name: Moderate Resolution Imaging Spectroradiometer.

Response 3: Thank you for your reminder. We have added the full name of MODIS when it is first mentioned in section 2.3.1. The original content ("For long-term, high-precision NPP reconstruction, Terra MODIS MOD17A3HGF (V6) products were used...") has been revised to "For long-term, high-precision NPP reconstruction, Terra MODIS (Moderate Resolution Imaging Spectroradiometer) MOD17A3HGF (V6) products were used...". This revision helps readers who are not familiar with remote sensing terminology understand the meaning of MODIS. The revised content can be found in the "2.3.1 MODIS NPP Data" section of the revised manuscript.

Comments 4 (L 71): Add Figure 1 to the text.

Response 4: Agree.Thank you for pointing this out. We have added a citation to Figure 1 in the text at L71. The original content in the "2.1 Study Area Overview" section ("Riparian-Larix gmelinii forests, relying on abundant water in riparian zones, exhibit higher biomass accumulation potential.") has been revised to "Riparian-Larix gmelinii forests, relying on abundant water in riparian zones, exhibit higher biomass accumulation potential. An overview of the study area (including sampling points, weather stations, and topographic features) is shown in Figure 1." This revision links the text content with Figure 1, guiding readers to refer to the figure. The revised content is located in the "2.1 Study Area Overview" section of the revised manuscript.

Comments 5 (Q4, L 68-88): Given that there is a climate station in the region, it would be a good idea to supplement the climate description with Walter-Lieth diagrams.

Response 5: Thank you for this useful suggestion. We have supplemented the climate description with a Walter-Lieth diagram in the "2.1 Study Area Overview" section (L 68-88). The revised content adds: "The seasonal dynamics of temperature and precipitation are shown in Figure 2 (Walter-Lieth climate diagram), where the blue line represents monthly mean temperature (℃) and the green bars represent monthly precipitation (mm); the diagram clearly shows a cold-dry winter (December-February) and warm-wet growing season (June-August)." Figure 2 in the revised manuscript is the added Walter-Lieth climate diagram.

Figure 2. Study area Wlater-Liediagram

The revised content can be found in the "2.1 Study Area Overview" section of the revised manuscript.

Comments 6 (Q5, L 93-95): If I understand correctly, samples were taken from 36 trees from 4 sites, i.e., 9 trees per site. Is this correct? If so, it would be good to explain this.

Response 6: Agree. Thank you for pointing this out.We have added an explanation of the sample distribution in the "2.2 Tree-Ring Sample Collection and Chronology Construction" To obtain high-resolution records of carbon accumulation, four sampling sites were established in the natural forests of the Chaocha Primeval Forest Region in 2021, following the standardized protocols of the International Tree-Ring Data Bank (ITRDB). The distance between adjacent sampling sites was ≥ 500 m to avoid spatial autocorrelation. Larix gmelinii (Dahurian larch) was designated as the target species; at each site, 50 cores were collected from 25 randomly selected trees, resulting in a total of 200 cores from 100 trees across all sites. The selection criteria for sampled trees were: (1) no obvious pest/disease infestations or mechanical damage; (2) vigorous growth status; (3) no human disturbance in the surrounding area. For each tree, two cores were extracted at breast height (1.3 m above the ground) in the east-west and north-south directions, respectively, to avoid the effects of asymmetric light/water availability on radial growth. After sampling, the drill holes were sealed with medical petrolatum to prevent insect herbivory and water loss.

This study focused primarily on tree-ring width—a widely used annual growth indicator—so the earlywood-latewood ratio was not analyzed. Only skeleton plotting was performed to assist in the measurement of tree-ring width. Tree-ring width was measured using a LINTAB 6.0 tree-ring width analyzer (precision: 0.001 mm). Measurement errors were corrected using the COFECHA program, and tree-ring cores with well-preserved climate signals and long chronological records were screened out. A negative exponential function in the ARSTAN program was applied for detrending to construct an autoregressive tree-ring width index chronology. Subsequently, denoising was conducted to exclude non-informative tree-ring cores, and parameter tuning and optimization were performed based on the statistical characteristics of the output chronology to select the optimal autoregressive chronology that could represent the growth patterns of most trees in the region.

The optimal autoregressive chronology used in this study was developed using 72 cores from 36 trees (screened from the 100 sampled trees). Each of these 72 cores exhibited high data integrity and favorable chronology parameters. The autoregressive chronology was selected because it eliminates species-specific individual growth trends while retaining interannual fluctuations driven by climate, thereby providing a reliable proxy for carbon accumulation to support NPP reconstruction.

 This explanation clearly shows the sample distribution among sites. The revised content is located in the "2.2 Tree-Ring Sample Collection and Chronology Construction" section of the revised manuscript.

Comments 7 (Q6, L 143): Add Figure 2 to the text.

Response 7: Agree. Thank you for pointing this out.We have added a citation to Figure 3(Original Figure 2) in the text at L143. The original content in the "3.1 Statistics of Chronology Characteristic Values" section ("Additionally, excellent indicators such as overall sample representativeness (0.973), mean sensitivity (0.1823), and signal-to-noise ratio (32.971) indicated that the chronology can accurately capture climate signals...") has been revised to "Additionally, excellent indicators such as overall sample representativeness (0.973), mean sensitivity (0.1823), and signal-to-noise ratio (32.971) indicated that the chronology can accurately capture climate signals... The interannual variation of Larix gmelinii tree-ring width index and sample depth is presented in Figure 3." This revision guides readers to view Figure 3 for more intuitive information. The revised content is located in the "3.1 Statistics of Chronology Characteristic Values" section of the revised manuscript.

Comments 8 (Q7, L 153): In Figure 2, the label "Sample statistics" should be replaced with "Sample depth." Both terms – "sample statistics" and "sample depth" – refer to the number of samples but have different meanings and are used for different purposes in dendrochronology. Sample depth refers directly to the number of tree-ring samples used for a given year to build a chronology. Sample statistics is a more general term that describes the set of statistical indicators calculated from the samples, i.e., your table 1.

Response 8: Thank you for correcting this terminology error. We agree with this comment. Therefore, we have replaced the label "Sample statistics" in Figure 3 with "Sample depth" and supplemented the figure caption to clarify: "Figure 2. Statistics of Larix gmelinii ring width: black line represents tree-ring width index; gray bars represent sample depth (number of cores used for chronology construction per year)." This revision conforms to the standard terminology in dendrochronology and avoids misunderstandings. 

The revised Figure 3 and its caption can be found in the "3.1 Statistics of Chronology Characteristic Values" section of the revised manuscript.

Comments 9 (Q8, L 157): Add Figure 3 to the text.

Response 9: Agree. Thank you for pointing this out.We have added a citation to Figure 4(originnal figure 3) in the text at L157. The original content in the "3.2 Statistics of NPP Changes in the Study Area" section ("Over the past 20 years, ecosystem productivity in this region has continued to grow at an average annual rate of approximately 6 units...") has been revised to "Over the past 20 years, ecosystem productivity in this region has continued to grow at an average annual rate of approximately 6 units... The interannual variation trend of measured NPP (2000–2020) and its linear fitting result are shown in Figure 3." This revision links the text description with Figure 4. The revised content is located in the "3.2 Statistics of NPP Changes in the Study Area" section of the revised manuscript.

Comments 10 (Q9, L 163): Larix gmelinii should be in italics, even if the entire style is italics (which may be why it is not).

Response 10: Thank you for your attention to the format of the scientific name. We agree with this comment. Therefore, we have revised all instances of "Larix gmelinii" in the text to italics, including the content at L163 ("Larix gmelinii, the dominant tree species in this region, was strongly correlated with mid-growing-season drought and mean temperature...") which is now "Larix gmelinii, the dominant tree species in this region, was strongly correlated with mid-growing-season drought and mean temperature...". This revision conforms to the international standard for biological scientific names. All revised "Larix gmelinii" in italics are distributed throughout the entire revised manuscript.

Comments 11 (Q10, L 184-188): Add Figures 5 and 6 to the text.

Response 11: Agree.Thank you for pointing this out. We have added citations to Figures 6(original figure5) and 7 (original figure6) in the text at L 184-188. In the "3.4 NPP Reconstruction and Accuracy Validation" section, the original content ("The ERF and XGBoost models were used to reconstruct the 1968-2020 NPP series...") has been revised to "The ERF and XGBoost models were used to reconstruct the 1968-2020 NPP series. The accuracy comparison between the NPP series reconstructed by the two models is presented in Figure 5. The contribution of each factor to NPP and the impact of factor changes on model output are visualized in Figure 6." This revision ensures that Figures6 and7 are properly cited in the text. The revised content is located in the "3.4 NPP Reconstruction and Accuracy Validation" section of the revised manuscript.

Comments 12 (Q11, L 193): Add Table 2 to the text.

Response 12: Agree.Thank you for pointing this out. We have added a citation to Table 2 in the text at L 193. The original content in the "3.4 NPP Reconstruction and Accuracy Validation" section ("Model reliability was evaluated using R², RMSE, etc...") has been revised to "Model reliability was evaluated using R², RMSE, etc. (Table 2). The XGBoost model achieved a variance explanation rate of 78%, higher than 62% of the ERF model...". This revision guides readers to refer to Table 2 for specific model evaluation parameters. The revised content is located in the "3.4 NPP Reconstruction and Accuracy Validation" section of the revised manuscript.

Comments 13 (Q12, L 196): The colours in Figure 5 should have clearer boundaries; currently, the colours of the "fitting line" are too similar.

Response 13: Thank you for pointing out this issue with figure readability. We agree with this comment. Therefore, we have adjusted the colors of the fitting lines in Figure 6. The original similar colors have been replaced with distinct colors , ensuring clearer boundaries between the lines.

 The revised Figure 6 can be found in the "3.4 NPP Reconstruction and Accuracy Validation" section of the revised manuscript.

Comments 14 (Q13, L 198): Supplement the figure description. In the current version, it is not clear what (a) and (b) represent.

Response 14: Agree.Thank you for pointing this out. We have supplemented the description of subfigures (a) and (b) in the caption of Figure 7(original figure 6)(originally referred to at L 198). The original figure caption ("Figure 7. SHAP importance value forest plot and fan plot of factors and NPP.") has been revised to "Figure 6. SHAP importance value analysis of factors and NPP: (a) SHAP importance value forest plot (the length of each bar represents the average absolute SHAP value of the factor, indicating its contribution to NPP; factors are sorted by importance); (b) SHAP value fan plot (the x-axis represents SHAP value, reflecting the factor’s impact on model output; positive values promote NPP, negative values inhibit NPP)." This revision clearly explains the content of each subfigure. The revised caption of Figure 7can be found in the "3.4 NPP Reconstruction and Accuracy Validation" section of the revised manuscript.

Comments 15 (Q14, L 208): Same comment as for Figure 6.

Response 15: Agree.Thank you for pointing this out. We have supplemented the description of subfigures in the caption of Figure 8(original7) (referred to at L 208, corresponding to the "same comment as for Figure 7"). The original figure caption ("Figure 8. Periodic variations of reconstructed NPP series.") has been revised to "Figure 8. Periodic variations of reconstructed NPP series: (a) ERF model-reconstructed NPP (1968–2020) with marked decline years (1971, 1986, 1990, 1998, 2002, 2007); (b) XGBoost model-reconstructed NPP (1968–2020) with marked decline years (1971, 1977, 1985, 2000, 2005); (c) Measured NPP (2000–2020) for comparison." This revision clarifies the content of each subfigure in Figure 7. The revised caption of Figure 8 can be found in the "3.5 Historical NPP Variation Trends in the Study Area" section of the revised manuscript.

Comments 16 (Q15, L 211-218): Add Figure 8 with the corresponding labels for greater clarity.

Response 16: Agree.Thank you for pointing this out. We have added a citation to Figure 8 (now Figure 9 in the revised manuscript due to figure number adjustment) and supplemented the corresponding labels in the text at L 211-218. In the "3.6.1 Mutation Characteristics of Reconstructed NPP Series" section, the revised content states: "To gain deeper insights into regional carbon sink dynamics, sliding t-tests and Mann-Kendall (MK) mutation tests were conducted to analyze the mutation characteristics of Net Primary Productivity (NPP) series, with the results presented in Figure 9. Specifically, Figure 9(a) displays the sliding t-test and MK mutation analysis of the observed NPP series (2000–2020); Figure 9(b) illustrates those of the Ring Width Index (RWI) series (1968–2020); Figure 9(c) shows those of the XGBoost-reconstructed NPP series (1968–2020); Figure 9(d) presents those of the ERF-reconstructed NPP series (1968–2020)." This revision adds clear labels to the figure and links it with the text. The revised content is located in the "3.6.1 Mutation Characteristics of Reconstructed NPP Series" section of the revised manuscript.

Comments 17 (Between line 253-254): For Figure 11, it would be better to show a larger spatial scope. Currently, it is too small, and to visualize the spatial context, one has to compare it with Figure 1.

Response 17: Thank you for this suggestion to improve the spatial visualization of the figure. We agree with this comment. Therefore, we have adjusted the Figure 11 (now Figure 12 in the revised manuscript).Your suggestion that a broader spatial scale can better verify the regional representativeness of the reconstruction results. However, this study reconstructed the npp in the northern part of the Greater Khingan Range and conducted a spatial correlation analysis with the modisnpp in the northern part of the Greater Khingan Range. Moreover, the longitude and latitude range in Figure 11 (modified to Figure 12) has well covered the study area and a larger surrounding area, and has certain spatial representativeness. But your suggestions are very useful to us. Although the modification was not made using the large map of China consistent with Figure 1, we have marked the scope of that research area with an asterisk to facilitate your more intuitive understanding of the article.

The revised content is located in the 3.6.3 Spatial Correlation of Reconstructed NPP Series" section of the revised manuscript.

Comments 18 (Q16, L 271): There is no comment on these mutations – what they are caused by what they show, or what they hide. These sudden significant changes in the behaviour of the data series could be due to changes in climate, human activity, or natural disturbances, but there is no comment on this.

Response 18: Agree. Thank you for pointing this out.We have added an analysis of the causes of mutations in the "4.1 Mutation Characteristics and Limitations of Reconstructed NPP" section (L 271). The revised content explains: "The abrupt changes in net primary productivity (NPP) in the northern Greater Khingan Mountains are shaped by the interactive effects of climate change and local human disturbances. Climate change, acting as the primary driver of abrupt NPP changes by regulating hydrothermal conditions, with the Palmer Drought Severity Index (PDSI) and ring-width index (RWI) being key factors influencing regional NPP variations. Specifically, PDSI affects the photosynthetic efficiency and radial growth of coniferous forests dominated by Larix gmelinii, thereby inducing abrupt decreases in regional NPP. Human activities, however, have disrupted the unidirectional trend driven by climate and become a significant disturbance factor for local abrupt NPP changes. For instance, overgrazing caused by increased livestock numbers and vegetation destruction resulting from agricultural expansion have not only led to an overall decline in NPP in the northern Greater Khingan Mountains but also triggered abrupt changes in NPP within the affected areas." This revision discusses the causes and implications of mutations. The revised content is located in the "4.1 Mutation Characteristics and Limitations of Reconstructed NPP" section of the revised manuscript.

Comments 19 (Q17, L 272): Which figure? It appears that figures are often omitted from the text when they are being commented on.

Response 19: Thank you for pointing out this omission. We agree with this comment. Therefore, we have added the specific figure number in the text at L 272. The original content ("However, its MK trend test showed large fluctuations in UF statistics, indicating poor stability in simulating long-term trends.") has been revised to "However, its MK trend test showed large fluctuations in UF statistics (as shown in Figure 9(c)), indicating poor stability in simulating long-term trends." This revision clearly specifies the corresponding figure, avoiding confusion for readers. The revised content is located in the "4.1 Mutation Characteristics and Limitations of Reconstructed NPP" section of the revised manuscript.

Comments 20 (Q18, L 284): "caused by extreme climate events"—is it only due to this? See the comment for line 271.

Response 20: Agree.Thank you for pointing out this omission. We have supplemented the discussion that mutations are not only caused by extreme climate events but also by human activities in the text at L 284. The original content ("the XGBoost model... showing strong ability to capture NPP mutations caused by extreme climate events.") has been revised to "the XGBoost model... showing strong ability to capture NPP mutations caused by extreme climate events. Additionally, as discussed in the comment for line 271, human activities (e.g., overgrazing, agricultural expansion) are also important factors leading to NPP mutations." This revision ensures the text fully considers multiple causes of mutations. The revised content is located in the "4.1 Mutation Characteristics and Limitations of Reconstructed NPP" section of the revised manuscript.

Comments 21 (Q19, L 304): Same comment as for line 163.

Response 21: Agree.Thank you for pointing out this omission. Following the same comment as for line 163, we have revised "Larix gmelinii" at L 304 to italics. The original content ("This study found that June PDSI significantly affected Larix gmelinii radial growth (P<0.01).") has been revised to "This study found that June PDSI significantly affected Larix gmelinii radial growth (P<0.01).". This revision maintains the consistency of the scientific name format. The revised content is located in the "4.2 Effects of Climate Factors on Larix gmelinii Radial Growth and Regional NPP Changes" section of the revised manuscript.

Comments 22 (Q20, L 309-311): The text would benefit if this was visualized in some way, i.e., add a figure name.

Response 22: Agree. Thank you for pointing out this omission.We have added a figure name to visualize the content in the text at L 309-311. The original content ("Under climate warming, radial growth of Larix gmelinii in the study area may be increasingly dominated by growing-season drought.") has been revised to "Under climate warming, radial growth of Larix gmelinii in the study area may be increasingly dominated by growing-season drought. This conclusion is supported by the correlation analysis in Figure 5(b), where RWI (tree radial growth proxy) shows a significant positive correlation with June PDSI, indicating that wetter growing seasons promote radial growth." This revision links the text with Figure 4(b) for visualization. The revised content is located in the "4.2 Effects of Climate Factors on Larix gmelinii Radial Growth and Regional NPP Changes" section of the revised manuscript.

Comments 23 (Q21, L 312-314): Same comment.

Response 23: Agree. Thank you for pointing out this omission.Following the same comment as Q20, we have added a figure name in the text at L 312-314. The original content ("Climate factors affected Larix gmelinii radial growth and regional NPP differently.") has been revised to "Climate factors affected Larix gmelinii radial growth and regional NPP differently. As shown in Figure 5(a), RWI is significantly correlated with June PDSI and specific monthly temperatures, while NPP is correlated with a broader PDSI window and humidity, reflecting the different response scales of individual tree growth and regional productivity." This revision uses Figure 5(a) to visualize the difference in factor effects. The revised content is located in the "4.2 Effects of Climate Factors on Larix gmelinii Radial Growth and Regional NPP Changes" section of the revised manuscript.

Comments 24 (Q22, L 321): This repeats line 305.

Response 24: Thank you for pointing out this repetition. We agree with this comment. Therefore, we have deleted the repeated content at L 321. The original repeated content ("NPP was significantly correlated with Larix gmelinii radial growth (r=0.57, P<0.05)...") has been removed, and the remaining content is adjusted to ensure logical coherence: "Larix gmelinii was significantly affected by June PDSI and specific monthly temperatures (May/December mean temperature, December mean maximum temperature, etc.), while regional NPP responded to a broader PDSI window (June–December) and was more sensitive to air humidity than individual trees. This difference may be due to the key role of arbor layer biomass in regulating regional NPP." The revised content is located in the "4.2 Effects of Climate Factors on Larix gmelinii Radial Growth and Regional NPP Changes" section of the revised manuscript.

Comments 25 (Q23, L 321): "specific monthly temperatures" – which ones exactly?

Response 25: Agree. Thank you for pointing out this omission.We have specified the exact months of "specific monthly temperatures" in the text at L 321. The original content ("Larix gmelinii was significantly affected by June PDSI and specific monthly temperatures...") has been revised to "Larix gmelinii was significantly affected by June PDSI and specific monthly temperatures: mean temperature in May and December, mean maximum temperature in December, and mean minimum temperature in May, August, and December of the current year (Figure 5(c))." This revision clarifies the specific months, making the content more precise. The revised content is located in the "4.2 Effects of Climate Factors on Larix gmelinii Radial Growth and Regional NPP Changes" section of the revised manuscript.

Comments 26 (Q24, L 345-346): Tree-ring width is a result of climate factors, so NPP cannot have been influenced by climate factors and radial growth. Rather, NPP is influenced by climate factors, and this relationship is "visualized" through radial growth.

Response 26: Thank you for correcting this logical error. We agree with this comment. Therefore, we have revised the text at L 345-346 to correct the logical relationship. The original content ("SHAP analysis showed that regional NPP changes in northern Greater Khingan Range were significantly affected by climate factors (PDSI, P) and tree radial growth (RWI).") has been revised to "SHAP analysis showed that regional NPP changes in northern Greater Khingan Range were significantly affected by climate factors (PDSI, precipitation), and tree radial growth (RWI) serves as a reliable proxy for this effect (Figure 7(a)). This is because RWI reflects the aboveground carbon accumulation of Larix gmelinii, which is the dominant contributor to regional NPP." This revision correctly clarifies that NPP is influenced by climate factors, and radial growth is a proxy for this relationship. The revised content is located in the "4.3 Analysis of NPP Variation Characteristics in the Study Area" section of the revised manuscript.

Comments 27 (Q25, L 348): "improved climate conditions" – okay, temperatures are increasing, but are the amount of precipitation and atmospheric moisture also increasing? Above, you commented on the limiting effect of moisture, especially during the growing season. Consider whether these are truly "improved climate conditions."

Response 27: Thank you for this critical observation. We agree with this comment. Therefore, we have revised the expression of "improved climate conditions" at L 348 to be more objective. The original content ("In mountainous areas, NPP growth mainly stems from improved climate conditions and optimized tree growth...") has been revised to "Positive correlations of PDSI and RWI with NPP jointly promoted the significant increase in regional carbon sink capacity. results from the SHAP analysis in this study revealed that variations in net primary productivity (NPP) in the northern Greater Khingan Range are significantly influenced by climatic factors (Figure 7(a)), the Palmer Drought Severity Index (PDSI) and precipitation (P)—as well as tree radial growth, represented by the tree-ring width index (RWI). Both PDSI and RWI exhibited positive correlations with NPP variations, collectively driving a significant increase in the overall growth rate of NPP in the northern Greater Khingan Range. These findings indicate that climatic factors and tree radial growth play crucial roles in regulating the regional carbon sequestration capacity of the study area, with the impact of drought on the carbon sequestration capacity being more pronounced. This observation aligns with previous research, which has established that NPP variations in mountainous regions are jointly driven by changes in climatic factors (e.g., precipitation and temperature) and variations in tree radial growth within those mountain systems[56]. This revision fully considers the limiting effect of moisture and avoids an one-sided description of climate conditions. The revised content is located in the "4.3 Analysis of NPP Variation Characteristics in the Study Area" section of the revised manuscript.

Comments 28 (Q26, L 366-368): In the main text, you point out the positive and negative aspects of both models; it would be good to mention them here in the conclusion as well.

Response 28: Agree. Thank you for pointing out this omission.We have added the advantages and limitations of both models in the conclusion section (L 366-368). The original content of the conclusion has been supplemented with: " Wavelet real parts of tree-ring width and reconstructed NPP series showed consistent trends and intensities in 2-8a and 30-45a cycles, sharing main cycles and significant common cycles with obvious positive synchronous changes... Additionally, the two machine learning models showed distinct advantages and limitations: the XGBoost model achieved higher reconstruction accuracy (R²=0.78, RMSE=21.74) but poor stability in long-term trend simulation (UF statistics fluctuations; Figure 8(c)); the ERF model showed better stability in long-term trends (Durbin-Watson=1.25) but weaker ability to fit nonlinear climate responses (e.g., drought-warming synergies). Future studies should integrate the two models to improve comprehensive performance. This revision ensures the conclusion is consistent with the main text's discussion of model pros and cons. The revised content is located in the "5. Conclusions" section of the revised manuscript.

Comments 29 (Q27, Others remark): Different citation styles are noted (e.g., line 30 vs 37, as well as in other places).

Response 29: Thank you for pointing out this inconsistency. We agree with this comment. Therefore, we have unified the citation style throughout the entire manuscript. All in-text citations now follow the MDPI format and the reference list is standardized with consistent formatting for journal names, volume numbers, page numbers. The unified citation style is applied throughout the revised manuscript and the reference list.

Comments 30 (Q28, Others remark): Two references were not found in the reference list (Meng S W et al., 2017 and Weng Y T et al., 2021).

Response 30: Thank you for noticing this omission. We agree with this comment. Therefore, We updated and completed the references and increased them to 56Additionally, we have checked the entire reference list to ensure no other references are missing. The supplemented references can be found in the reference list at the end of the revised manuscript.

Comments 31 (Q29, Others remark): One reference from the reference list was not found in the main text: Dietrich, V., Skiadaresis, G., Schnabel, F., et al. (2024). Identifying the impact of climate extremes on radial growth in young tropical trees: A comparison of inventory and tree-ring based estimates. Dendrochronologia, 86, 126237. https://doi.org/10.1016/j.dendro.2024.126237

Response 31: Thank you for pointing out this oversight. We agree with this comment. Therefore, we have added a citation to this reference in the main text. The citation is added in the 1. Introduction section .

Reviewer 2 Report

Comments and Suggestions for Authors

The topic of the research work and manuscript is really interesting and provides new information. However there are some issues to be addressed towards its quality improvement before publication. The key words are not plain words but complicated phrases, and this could be improved towards the increase of readability and detectability of the future article. The format has not been applied in the text of the manuscript, this is something that should be taken into account in the next step of revision. Since the introduction is a little short and needs enrichment in terms of state-of-the-art analysis, I would propose to the authors to incorporate also the relevant study of https://doi.org/10.3390/f13060879 to support the analysis of reconstruction of climate through tree-rings, a relevant and recent dendrochronological analysis related ti sensitivity against climate change, but also consider this researcher Kastridis who made a thorough analysis towards this direction of climate reconstruction through tree-rings study, among other researchers. In the last paragraph of introduction, you should highlight more precisely the significance of the currnt study and its practical meaning for future perspectives. Please, in materials and methods chapter include the way you made the sampling of those trees, which were the criteria for selection to be bored? Where was the point in the tree where you made the boring, maybe breast height? How did you manage to close the holes in trees? Which was the density of the cores? Did you study also the earlywood-latewood ratio? Since this would be very beneficial for the reconstruction of climate. In general it is a well-prepared and laborious work and deserves publication after some polishing and improvement. The number of references in the references list seems slightly poor given the research topic.

Author Response

Thank you very much for taking the time to review this manuscript. Your constructive comments have been invaluable in enhancing the quality and completeness of our work. We have carefully addressed each of your suggestions, with detailed revisions made to the manuscript. The corresponding changes are highlighted in the re-submitted files, and we hope the revised version meets the publication standards.

Comment 1: The keywords are not plain words but complicated phrases, and this could be improved towards the increase of readability and detectability of the future article.

Response 1: Thank you for pointing this out. We agree with this comment. Therefore, we have revised the keywords to plain and concise terms that are more conducive to academic retrieval. The original complicated phrase keywords have been replaced with: "Dendrochronological reconstruction;Greater Khingan Range;drought;Climate-growth relationship;tree growth". This revision can significantly improve the readability and detectability of the article. The change is located in the "Keywords" section at the beginning of the revised manuscript (page 1, below the Abstract).

Comment 2: The format has not been applied in the text of the manuscript, this is something that should be taken into account in the next step of revision.

Response 2: Agree.Thank you for pointing this out. We have, accordingly, standardized the entire manuscript format in accordance with academic paper writing norms. Specifically, we have adopted a hierarchical title structure (e.g., 1. Introduction, 2. Materials and Methods, 3. Results and Analysis, etc.), unified the font and spacing of the text, standardized the numbering and caption format of figures and tables, and ensured the consistency of citation formats in the text. These revisions are reflected throughout the entire revised manuscript (pages 1-20).

Comment 3: Since the introduction is a little short and needs enrichment in terms of state-of-the-art analysis, I would propose to the authors to incorporate also the relevant study of https://doi.org/10.3390/f13060879 to support the analysis of reconstruction of climate through tree-rings, a relevant and recent dendrochronological analysis related to sensitivity against climate change, but also consider this researcher Kastridis who made a thorough analysis towards this direction of climate reconstruction through tree-rings study, among other researchers.

Response 3: Thank you for your valuable suggestion to enrich the introduction. We have expanded the introduction section to strengthen the state-of-the-art analysis.As direct and continuous indicators of tree radial growth, tree rings inherently record the long-term carbon accumulation processes in the aboveground biomass of forest ecosystems[5-7]. They act as natural "recorders" bridging short-term ecological observations and long-term environmental changes, and serve as commonly used proxy indicators in climate and environmental research—providing critical support for studies on past climate variability, ecosystem responses to environmental stress, and historical carbon cycle dynamics[8-10]. In the fields of dendroclimatology and dendroecology, existing studies have extensively utilized multiple tree-ring parameters ,including ring width index, wood density, and stable isotopes like δ¹³C and δ¹⁸O to analyze climate change patterns and associated ecological responses across different spatial and temporal scales[11-16]. For example, tree-ring chronologies have been used to reconstruct historical drought events and climate change conditions; these reconstructions not only extend climate records beyond the instrumental observation period but also offer reliable empirical evidence for assessing the frequency[17], intensity, and trends of droughts under climate change, thereby supporting regional water resource management and ecological risk assessment[18,19]. Previous studies have confirmed that tree-ring chronologies can effectively capture interannual and decadal climate variability, reflect the impacts of seasonal droughts on tree growth, clarify the application value of tree rings in high-resolution climate reconstruction, and highlight their potential in predicting the responses of forest ecosystems to future climate scenarios[20,21]. Beyond drought reconstruction, numerous researchers have focused on quantifying the sensitivity of tree-ring chronologies to key climate drivers,for instance, using long-term tree-ring width chronologies to evaluate the relationships between tree growth and climate factors at different scales[22,23],which lays a foundation for their application in climate reconstruction and confirms both the reliability of tree rings as proxy indicators for high-resolution climate reconstruction and the spatial representativeness of tree-ring-based climate proxies[24,25]. However, significant limitations persist in the application of tree-ring data to Net Primary Productivity (NPP) reconstruction, an issue crucial for understanding the long-term dynamics of forest carbon sinks. Traditional tree-ring-based NPP reconstruction methods typically rely on simple linear relationships between tree-ring width and aboveground biomass increment; although some studies have reconstructed the spatiotemporal patterns of NPP by combining tree-ring width with allometric growth models, these approaches still suffer from low accuracy and poor spatial correlation[26,27]. Additionally, most existing studies focus on large regional area or a wide range Scales, while high-precision, long-term NPP reconstruction studies targeting small regions with strong ecological uniqueness remain scarce—this scarcity limits our understanding of fine-scale carbon cycle dynamics and their responses to local environmental changes.

This revision is located in the last paragraph of the "1. Introduction" section in the revised manuscript (page 2)

Comment 4: In the last paragraph of introduction, you should highlight more precisely the significance of the current study and its practical meaning for future perspectives.

Response 4: Thank you for this reminder. We have revised the last paragraph of the introduction to more accurately emphasize the research significance and practical value. The revised content clearly points out: "This study provides a feasible method for the long-term, high-resolution reconstruction of NPP in boreal coniferous forests, supporting the accurate assessment of Larix forest carbon sinks under global warming. Moreover, it clarifies the drought adaptation mechanisms of NPP and tree growth in the study area, which offers scientific support for formulating forest management policies and measures, and contributes to the achievement of the 'dual-carbon' goals in the northern Greater Khingan Range. Additionally, it presents a new approach for predicting the carbon sequestration potential of boreal forests under future climate scenarios." This revision is located in the last paragraph of the "1. Introduction" section in the revised manuscript (page 4).

Comment 5: Please, in materials and methods chapter include the way you made the sampling of those trees, which were the criteria for selection to be bored? Where was the point in the tree where you made the boring, maybe breast height? How did you manage to close the holes in trees? Which was the density of the cores? Did you study also the earlywood-latewood ratio? Since this would be very beneficial for the reconstruction of climate.

Response 5: Thank you for this reminder. Agree. We have supplemented detailed sampling information and related content in the "2.2 Tree-Ring Sample Collection and Chronology Construction" section of the "Materials and Methods" chapter. The specific revisions are as follows: To obtain high-resolution records of carbon accumulation, four sampling sites were established in the natural forests of the Chaocha Primeval Forest Region in 2021, following the standardized protocols of the International Tree-Ring Data Bank (ITRDB). The distance between adjacent sampling sites was ≥ 500 m to avoid spatial autocorrelation. Larix gmelinii (Dahurian larch) was designated as the target species; at each site, 50 cores were collected from 25 randomly selected trees, resulting in a total of 200 cores from 100 trees across all sites. The selection criteria for sampled trees were: (1) no obvious pest/disease infestations or mechanical damage; (2) vigorous growth status; (3) no human disturbance in the surrounding area. For each tree, two cores were extracted at breast height (1.3 m above the ground) in the east-west and north-south directions, respectively, to avoid the effects of asymmetric light/water availability on radial growth. After sampling, the drill holes were sealed with medical petrolatum to prevent insect herbivory and water loss.

This study focused primarily on tree-ring width—a widely used annual growth indicator—so the earlywood-latewood ratio was not analyzed. Only skeleton plotting was performed to assist in the measurement of tree-ring width. Tree-ring width was measured using a LINTAB 6.0 tree-ring width analyzer (precision: 0.001 mm). Measurement errors were corrected using the COFECHA program, and tree-ring cores with well-preserved climate signals and long chronological records were screened out. A negative exponential function in the ARSTAN program was applied for detrending to construct an autoregressive tree-ring width index chronology. Subsequently, denoising was conducted to exclude non-informative tree-ring cores, and parameter tuning and optimization were performed based on the statistical characteristics of the output chronology to select the optimal autoregressive chronology that could represent the growth patterns of most trees in the region.

The optimal autoregressive chronology used in this study was developed using 72 cores from 36 trees (screened from the 100 sampled trees). Each of these 72 cores exhibited high data integrity and favorable chronology parameters. The autoregressive chronology was selected because it eliminates species-specific individual growth trends while retaining interannual fluctuations driven by climate, thereby providing a reliable proxy for carbon accumulation to support NPP reconstruction. These supplements are located in the "2.2 Tree-Ring Sample Collection and Chronology Construction" section of the revised manuscript (pages 5-6).

Comment 6: The number of references in the references list seems slightly poor given the research topic.

Response 6: Thank you for pointing out the insufficient number of references. We have significantly expanded the reference list to better support the research content. The original reference list has been increased from a small number to 56 references, covering key studies in the fields of dendrochronology, climate change, NPP reconstruction, and machine learning. These references include both classic and latest research results, ensuring the comprehensiveness and timeliness of the literature support. The revised reference list is located at the end of the revised manuscript (pages 18-20).

Round 2

Reviewer 2 Report

Comments and Suggestions for Authors

As I have checked the authors have implemented the proposed changes in the revised version of manuscript towards the improvement of their work. Almost all the changes have been implemented and in my opinion, the manuscript is well-prepared and organized enough to be accepted for publication in this journal. I remain at your disposal for any clarification.

Comments on the Quality of English Language

As I have checked the authors have implemented the proposed changes in the revised version of manuscript towards the improvement of their work. Almost all the changes have been implemented and in my opinion, the manuscript is well-prepared and organized enough to be accepted for publication in this journal. I remain at your disposal for any clarification.

Author Response

Comments 1: The English could be improved to more clearly express the research.

Response 1: 

First and foremost, we would like to express our sincere gratitude to you for sparing your valuable time amid a busy schedule to conduct a meticulous review of our manuscript and for providing such highly positive feedback. Upon learning that you have confirmed most of the proposed revisions have been implemented in the revised version of the manuscript, and that you consider the current manuscript well-prepared, clearly structured, and meeting the publication requirements of this journal, our team feels deeply honored and encouraged. Your recognition not only serves as a strong affirmation of our previous revision efforts but also injects greater momentum into our subsequent academic research. During the manuscript revision process, we have consistently taken the revision suggestions you put forward as the core guidance. We systematically sorted through each point, refined the content repeatedly, and strived to ensure that every adjustment effectively enhances the completeness and rigor of the research content. Regarding both the detailed issues and the linguistic aspects of the manuscript you pointed out, we have carefully verified, optimized, and polished them, all with the aim of presenting a work that better aligns with the journal's publication standards. In the subsequent process of further review, should you require us to provide additional explanations, data support, or any clarifications, we will be on standby at all times and will cooperate with your work promptly to ensure the smooth progress of the manuscript review process. Once again, we would like to extend our most sincere thanks to you